# Detection of Feline Coronavirus RNA in Cats with Feline Infectious Peritonitis and Their Housemates

**DOI:** 10.3390/v17070948

**Published:** 2025-07-04

**Authors:** Phoenix M. Shepherd, Amy Elbe, Brianna M. Lynch, Erin Lashnits, Robert N. Kirchdoerfer

**Affiliations:** 1Biochemistry Department, University of Wisconsin-Madison, Madison, WI 53706, USA; 2Institute for Molecular Virology, University of Wisconsin-Madison, Madison, WI 53706, USA; 3Center for Quantitative Cell Imaging, University of Wisconsin-Madison, Madison, WI 53706, USA; 4Department of Medical Sciences, School of Veterinary Medicine, University of Wisconsin-Madison, Madison, WI 53706, USA

**Keywords:** feline coronavirus, FECV, FIPV, feline infectious peritonitis

## Abstract

Feline coronavirus (FCoV), the causative agent behind feline infectious peritonitis (FIP), is one of the biggest infectious threats to feline health. Despite this threat, the tissue distribution and viral RNA levels in cats infected with feline coronaviruses are poorly understood in the context of natural infection. Here, we used a two-step reverse-transcription quantitative PCR (RT-qPCR) to examine viral RNA levels from different sampling sites in both cats that have been clinically suspected of FIP and their feline housemates. We show that the distribution and amount of FCoV viral RNA does not differ between FCoV-infected cats with FIP and their feline housemates in blood, conjunctiva, or feces. Furthermore, in all FIP and non-FIP cases, viral RNA levels were higher in fecal samples than the blood. Taken together, these results show that amount of viral RNA does not differ between FCoV-infected cats with FIP and their healthy housemates in several sample types. Our results indicate a need for closer examination of FCoV pathogenesis independent of viral dissemination, including an assessment of intrahost evolution of FCoVs and FCoVs’ interactions with the feline immune system.

## 1. Introduction

Coronaviruses are known for their ability to infect and subsequently cause disease in a wide variety of host species. Several human coronaviruses, such as MERS-CoV, SARS-CoV, and SARS-CoV-2, emerged due to spillover events from animal to human populations [1,2,3,4,5], likely because of promiscuous receptor use and due to the high rate at which these viruses mutate [6,7,8]. Feline coronaviruses are members of the *Alphacoronavirus* genus (family *Coronaviridae*, order *Nidovirales*) and are related to canine coronaviruses (CCoV) infecting dogs as well as transmissible gastroenteritis virus and porcine respiratory coronavirus (PRCV) infecting pigs [9]. Like their canine counterparts, FCoVs circulate in two serotypes. Type I is the most common serotype in the United States, accounting for as much as 80% of natural infections [10,11]. While less common, type II feline coronaviruses also circulate in the US [10,11,12,13]. FCoVs have remained an enigma in the scientific community due to their ability to cause both mild and severe diseases, an ability that is still poorly understood, yet is of paramount importance in developing antiviral and vaccination strategies [14,15].

FCoVs are infamous for their ability to either cause asymptomatic, benign, or severe disease in cats. In most cases, FCoV infections are entirely asymptomatic or cause mild gastrointestinal disease characterized by diarrhea or appetite loss [16,17,18]. Through unknown mechanisms, FCoVs may also cause a severe systemic disease known as feline infectious peritonitis (FIP).

It is hypothesized that FIP-associated FCoV is not transmissible from cat to cat, but rather is the result of a parental FCoV mutating over the course of infection to develop into FIP [14,15]. Generally, FCoVs are thought to more productively infect the cat’s intestinal epithelium, where the virus may establish a low, persistent infection that is shed in the feces. FIP-associated FCoVs are more likely to be associated with systemic infections and have the ability to infect macrophages [14,19,20,21,22]. The precise determinants of changes to FCoV tropism are currently under debate, but likely involve genetic mutations in the coronaviral spike protein [23]. Amino acid changes in the S1/S2 junction in feline coronavirus spike have been hypothesized to aid in FIP-associated viral dissemination by altering spike’s susceptibility to proteolytic cleavage [24]. Several studies have shown a methionine-to-leucine mutation at position 1058 (M1058L) and a serine-to-alanine mutation at position 1060 (S1060A) in the fusion subunit of spike in cases of FIP, but it is currently unknown how such changes may contribute to FIP [24,25,26]. Others have also reported the presence of FCoV RNA in systemic samples from healthy cats, suggesting that viral tropism changes are involved, but not solely responsible for the pathogenic change [16,27,28,29].

Another hypothesis for the pathogenesis of FIP is spike’s interaction with the feline immune system. Work has been performed in vitro with convalescent cat serum to assess antibody-dependent enhancement (ADE) of FCoV infections, in which non-neutralizing antibodies bind to FCoV virions and traffic the virus into macrophages [30,31]. ADE is thus hypothesized to cause systemic spread of FCoVs and lead to FIP.

FIP can manifest in many forms, some of which are neurotropic [32,33,34,35]. FIP is most commonly recognized by the presence of granulomatous lesions and inflammation in several organs, such as the liver, kidney, and spleen. In some cases, fibrinous, granulomatous serositis [33,34,35,36] occurs, which results in the formation of protein-rich effusions in body cavities of diseased cats and is termed “wet” FIP [35,36]. Cats afflicted with neurotropic “dry” FIP may experience seizures, ocular disease, behavioral changes, ataxia, and more [33,34,35,36]. Recently, an alarming outbreak of an FIP-causing FCoV in Cyprus has sparked scientific interest in understanding how FCoVs evolve into new pathotypes within hosts to cause severe disease [36].

The gold standard for detection and diagnosis of FIP is to use immunohistochemistry targeting the FCoV nucleocapsid or matrix proteins in fixed tissues from biopsies or necropsies [33,34]. The gold standard for detection of FCoV RNA extracted from diseased cats is RT-PCR. Treatment options for FIP are limited. Recently, nucleoside analogs GS-441524 and the related antiviral remdesivir, used to treat SARS-CoV-2 infections, have both been shown to be effective for curing cats of FIP with limited side effects [27,37,38,39]. In addition, an inhibitor of the FCoV viral protease, GC376, has also been shown as effective against viral replication in cats, albeit to a lesser extent [28,40,41]. While effective, these treatments are not preventative and tend to be expensive.

Numerous studies have been conducted to assess the amount of viral RNA present in feline samples, including blood, effusions, and feces using either quantitative or non-quantitative RT-PCR to elucidate FCoV tropism changes [14,16,25,33,34,35,36,37,38,39,40]. The level of viral RNA varies from study to study, generally showing low to unquantifiable levels of viral RNA in the blood but higher more persistent amounts of FCoV RNA in the feces [16,17,29,39,42,43,44,45,46,47]. Detection of FCoV RNA in body organs of diseased cats has also been assessed, with more detection of RNA in the intestines in the cat, particularly within the colon [16]. It is speculated that the colon of cats infected with FCoV may serve as a source of virus to be shed in the feces [16]. It is known that most cats infected with FCoV shed virus in their feces, sometimes up to months after initial infection [11,16,17]. Treatment with anti-FCoV drugs, such as GS-441524, has been shown to lessen the amount of virus shed in the feces [19]. Taken together, these studies show that in multi-cat environments, FCoV can be detected less reliably in the blood, but more so in the feces shed from both sick and healthy cats. Less work has been carried out to assess the amount of FCoV RNA in neurotropic FIP cases, but others have detected FCoV RNA in the cerebral spinal fluid and aqueous humor of the eye in FIP-diagnosed cats [18,25].

Greater insight into the prevalence of feline coronavirus across different sites within the cat are needed to develop better vaccination and treatment strategies. Progress is hindered by the lack of reliable methodologies to measure feline coronaviruses in a variety of samples within a single cat as well as a lack of matched controls between FCoV- and FIP-associated FCoV-infected cats. Here, we have developed a highly sensitive, two-step Taqman-probe-based RT-qPCR assay that targets the 3′ end of the feline coronavirus RNA genome to quantify viral RNA in sick and healthy cats infected with FCoVs. We have recruited a cohort of cats who have been clinically suspected of FIP or are housemates to those FIP cats, with the assumption that both groups may have experienced infection of the same parental virus, enabling us to compare the presence and amount of viral RNA between direct, matched controls. We use RT-qPCR in both symptomatic FIP cats and their housemates to better understand viral spread in both cases of severe FIP and asymptomatic FCoV infections.

## 2. Materials and Methods

### 2.1. Recruitment of Study Cats and Housemates

Cats (n = 75) were enrolled at the University of Wisconsin-Madison Veterinary Teaching Hospital, University of Wisconsin Veterinary Care after an evaluation of symptoms based on owner reports and veterinary reviews. Each cat was assigned an FIP Identification (FIPID) number to track sample intake and household relations. Cats were diagnosed according to the 2022 AAFP/EveryCat FIP diagnosis guidelines and the European Advisory Board on Cat Diseases FIP guidelines [48]. Definitive diagnosis of FIP based on immunohistochemistry was not required for inclusion; instead, cats could be included based on a clinical suspicion. If known, the date of birth, age at time of diagnosis, sex, breed, diagnosis category (FIP confirmed or suspected), and predominant clinical form were recorded at the time of enrollment. Medical records were reviewed by a board-certified small animal internal medicine specialist veterinarian (EL) to determine a diagnosis. Cats were not included if an alternative diagnosis (such as toxoplasmosis, congestive heart failure, or septic pleuritis) was reached. Cats that had already begun treatment with antiviral agents before the time of sampling were excluded from the analyses.

Blood, fecal and conjunctival swabs, and effusion (if present) from cases of probable FIP (n = 36) were taken and owners were asked if the cat had healthy feline housemates. Housemates (n = 39) were recruited at a later date, depending on the owners’ availabilities. Due to the nature of sampling from naturally infected, client-owned cats, we could not control for the time of FIP and housemate sampling. Housemate appointments were usually scheduled within a week of the initial FIP cat appointments. Nine unexposed, healthy cats from single-cat households were also recruited to assess the specificity of our assay. Due to the nature of this study, the infection histories of all cats used in this case study are unknown. All cats used in this study were enrolled with the owners’ consent, and studies were performed under IACUC approval protocol #V006485.

### 2.2. Feline Sample Processing

Blood Draws: We drew 2–3 mL of whole blood, depending on the cat weight and anemic status, from cats using a vacutainer blood draw system (Vacutainer Tubes 367856 and 387812, Becton Dickinson, Franklin Lakes, NJ, USA). Blood was drawn into 3 mL vacutainer tubes spray-coated with 5.4 mg of K_2_EDTA (whole blood). Whole blood from the K_2_EDTA-coated tubes was aliquoted prior to centrifugation at 1400× *g* for 15 min. The lower-density layer consisting of plasma was removed and spun again at 500× *g* for eight minutes to pellet any remaining cells. Whole blood and plasma were then aliquoted into sterile 1.5 mL microcentrifuge tubes and frozen at −80 °C.

Effusion: Total effusion volume (between 100 µL and 10 mL, depending on availability) was collected in a sterile 1.5 mL microcentrifuge tube(s). Effusion was spun at 500× *g* for eight minutes, and the supernatant was aliquoted into sterile 1.5 mL microcentrifuge tubes and frozen at −80 °C.

Fecal Swabs: Anal swabs (FLOQSwabs 220250, Copan, Murrieta, CA, USA) were used to collect fecal samples. Swabs were inserted dry and swirled on the swab axis for 10 s to collect feces. Fecal swabs were then placed into 500 µL of 1× DNA/RNA Shield (Zymo Research R1200, Irvine, CA, USA) and either frozen at −80 °C or immediately processed to collect RNA. Swabs were removed via sterilized forceps and wrung out on sides of the tubes. The 1× DNA/RNA Shield tube containing wrung-out fecal samples were relabeled with appropriate FIPIDs and frozen at −80 °C.

Conjunctival Swabs: Conjunctival swabs (FLOQSwabs 220250, Copan, Murrieta, CA, USA) were used to collect ocular samples. Cat conjunctiva was swabbed for 10 s and placed into 1× DNA/RNA Shield (Zymo Research R1200, Irvine, CA, USA) and processed as described with fecal swabs above.

### 2.3. Extraction of Feline Coronavirus RNA and Synthesis of cDNA

We used 100 µL of whole blood, plasma, effusion, conjunctival swab 1× DNA/RNA Shield, or fecal swab 1× DNA/RNA Shield for viral RNA extraction. Extraction was performed by using a Viral Quick RNA Extraction Kit (Zymo Research R1034, Irvine, CA, USA) and RNA Cleanup Kit (Zymo Research R1019, Irvine, CA, USA) following the protocols for DNase treatment. Prior to adding the Viral RNA Buffer, 2 µL of baker’s yeast (1 mg/mL) RNA (Millipore Sigma R6750 Madison, WI, USA) was added to every sample as carrier RNA. RNA was immediately used in reverse-transcription or stored at −80 °C. For FCoV cDNA synthesis, 10 µL of RNA was added to 6 µL of nuclease-free water and 4 uL of Maxima H-minus RT master mix (ThermoFisher EP0752, Waltham, MA, USA). Thermocycler conditions were 25 °C for 10 min, 65 °C for 15 min, 85 °C for 5 min, and a 10 °C hold. cDNA was stored at −20 °C until used in the RT-qPCR assay.

### 2.4. Taqman-Probe RT-qPCR

In vitro Transcription (IVT) of RNA Standards: 152-nucleotide sequences matching the conserved 3′ end of the FCoV, CCoV, or PRCV were synthesized as DNA in a pBluescript vector, containing a 5′ T7 promoter and 3′ restriction sites (GenScript, Piscataway, NJ, USA). DNA from all three plasmids were linearized via restriction digestion using the *Xbal* restriction enzyme, and IVT was conducted using NEB IVT buffer and T7 RNA polymerase (New England Biolabs M0251, Ipswitch, MA, USA). Standards were then treated with DNaseI (New England Biolabs M0303, Ipswitch, MA, USA) and an RNA Cleanup Kit (Zymo Research R1019, Irvine, CA, USA). RNA was quantified by UV absorbance (DeNovix Inc., Wilmington, DE, USA) and diluted for a standard curve. Standard curves were formed by serially diluting RNA stocks from 10^8^ copies of RNA/mL to 10^0^ copies of RNA/mL and reverse transcribed as described above.

Primers and Probes: To locate target sites to accommodate primer binding, we used ClustalOmega [49] software to align the genomes of 71 publicly available FCoV genomes. Primers and probes targeting a conserved region (accession number GQ152141.1 nucleotide position 28,646–28,775) in FCoV genomes were ordered through Integrated DNA Technologies (IDT). The forward sequence is CAACCCGATGTTTAAAACTGGT while the reverse sequence is CTAAATCTAGCATTGCCAAATCAAAT. The TaqMan probe forward sequence was manufactured by IDT and contains a 5′ 6-FAM fluorophore, a ZEN quencher, and a 3′ Iowa Black fluorescein quencher. The probe sequence is /56-FAM/CTACTCTTG/ZEN/TACAGAATGGTAAGCACGT/3IABkFQ/.

Quantitative PCR: We added 2 µL of sample cDNA or standard to a clear plastic 96-well plate (GeneMate, 490003-822, Rockville, MD, USA). Mastermix containing forward, reverse, and probe oligonucleotides (final concentration 1 µM), 2× iTaq Universal Probes Supermix (BioRad, 1725130, Hercules, CA, USA), and nuclease-free water was added to each well and mixed. The total reaction volume was 12.5 µL. A BioRad CFX qPCR machine was used to read the FAM fluorescence. The amplification conditions were as follows: initial denaturation 95 °C for 5 min followed by 40 cycles of denaturation at 95 °C for 10 s, annealing and elongation at 60 °C for 30 s, and detection of the 6-FAM fluorophore. RNA copies/mL were calculated based on the threshold count in each sample. All samples and standards were run in triplicate. Assays were accepted if the PCR efficiency was between 95% and 110% with a standard curve R-squared value of >0.99. Negative controls consisting of PCR mastermix, primers and a probe, and nuclease-free water with no template DNA were used to assess potential contamination. Limits of quantification (LoQ) and limits of detection (LoD) were determined by the BioRad CFX qPCR software, using 10% of the maximum relative fluorescence unit (RFU) of the least-dilute standard as a threshold for amplification. Each sample was categorized as quantifiable, detectable, or below the limit of detection, based on the number of cycles used to cross the threshold and the average RFU of the last five cycles. LoQ and LoD were based on the performance of the standards per assay. For these assays, the LoQ was typically between 700 and 1000 RNA copies/mL.

Statistics: Statistical analysis and descriptive statistics were performed using GraphPad Prism version 10.2.1 for MacOS (GraphPad Software, Boston, MA, USA). To compare unmatched FIP cats and their housemates, we used unpaired Student *t*-tests and defined statistical significance by a *p*-value of less than 0.05. To compare samples from matched cats of the same household, a paired *t*-test was used. All samples were run as technical triplicates, including standards. Each technical triplicate was then averaged and used in statistical analysis. Only quantifiable samples were used in statistical analysis.

## 3. Results

### 3.1. Assessing Specificity of the RT-qPCR Assay

To examine how specific our assay was at detecting feline coronaviruses, we designed additional standard curves, based on homologous sequences in two closely related alphacoronaviruses: canine coronavirus (CCoV) and porcine respiratory coronavirus (PRCV) (Appendix A). These three viruses are all considered to be the same species, *Alphacoronavirus suis*. Our forward, reverse, and probe sequences have no mismatches to PRCV 27,495–27,646 (GenBank KR270796.1) or CCoV 28,803–28,954 (GenBank PP526172.1). There was sufficient cross-reactivity of our FCoV primers with CCoV (cycle threshold values were approximately even), and to a lesser extent PRCV (which had delayed cycle threshold values). Efficiencies for both CCoV and PRCV curves, however, were lower than accepted FCoV curves (88.4% and 87.3% for CCoV and PRCV, respectively, and 102% for FCoV). The LoQ from the CCoV standard was the same as with FCoV (~1000 copies/mL) while the PRCV was ~100× higher. Due to primer and probe sequences being found in both CCoV and PRCV genomes, we cannot rule out that the cats presented in this study were infected with PRCV or CCoV rather than FCoV.

Next, we wanted to know if our assay is sensitive to background RNA or genomic DNA from feline hosts. We first isolated RNA from two different feline cell lines: Fcwf-4 macrophages (ATCC CRL-278) and AK-D lung epithelium (ATCC CCL-150). After RT-qPCR, both samples were below the limit of detection. We also recruited nine healthy housecats with unknown infection histories from single-cat households and sampled them using the methodology described above. Data displayed in Appendix A shows some quantifiable viral RNA in one or two cats from each sample (other than plasma, which had undetectable levels in all cats tested).

### 3.2. Sites and Levels of FCoV RNA Do Not Distinguish Clinical FIP

Plasma: To understand the systemic spread of FCoV in both cats with (n = 26) and without FIP (n = 42), we used our RT-qPCR assay to determine the total amount of viral RNA in cat plasma (Figure 1A). Results show that overall positivity in both groups was low (26.9% for FIP cats and 19% for healthy housemates), meaning most cats tested below the limit of detection in our assay. Those cats that did test above the limit of quantification tended to have low amounts of viral RNA in their plasma. When comparing diseased and non-diseased cats, we observed no significant difference in viral RNA levels between positive cats from the two groups (Figure 1A, *p* = 0.2134).

Whole Blood: Some studies have found that FCoV RNA is cell-associated, meaning acellular plasma may result in negative or low RT-qPCR signals [42,43,44,45,46,50]. We collected whole blood samples prior to centrifugation for RT-qPCR. Aggregated results show the same prevalence of positivity in both FIP (n = 17) and non-FIP cats (n = 23). Positivity was 35.3% for FIP cats and 26.1% for healthy housemates. However, average RNA levels in whole blood were slightly increased compared to plasma (Figure 1B), although the difference is not significant. When comparing RNA levels in whole blood samples of FIP and non-FIP cats, there was also no significant difference between the two groups, similar to the results for plasma samples (Figure 1B, *p* = 0.7657).

Fecal Swabs: To measure the amount of enteric FCoV RNA, we performed RT-qPCR on fecal swab samples. In contrast to the plasma and whole blood samples, we observed high amounts of viral RNA extracted from fecal swabs (Figure 1C). Additionally, both FIP (n = 27) and non-FIP (n = 38) cat groups presented greater overall positivity (74.1% for FIP cats and 73.7% for healthy housemates). We observed comparable amounts of viral RNA between FIP and non-FIP cats (Figure 1C, *p* = 0.2816). The percent of positivity and high RNA levels in both groups lead us to speculate that there may be viral shedding in both FIP cats and their feline housemates.

Conjunctival Swabs: Because some forms of feline coronaviruses can manifest as ocular disease [33,35], we sampled a portion of our cats both with FIP-associated FCoV infection (n = 14) and without (n = 14) for the presence of feline coronavirus in the conjunctiva using the same methodology as our fecal swabs. Surprisingly, most cats showed detectable levels of viral RNA in these samples (Table 1). Similar to other sampling sites, there was no statistically significant difference in FCoV RNA levels between cats with and without FIP disease (Figure 1D, *p* = 0.7681). Positivity was 64.3% for FIP cats and 71.4% for healthy housemates.

Effusion: We also used our RT-qPCR assay to measure the level of viral RNA in effusions from “wet” FIP cats (n = 16). Here, we observed high levels of viral RNA as well as a large percentage of overall positivity at 75% (Figure 1E, Table 1).

### 3.3. Viral RNA Levels Are Not Different Between FIP Cats and Their Housemates

Due to the lack of significant differences in FIP cats and their housemates in the overall cohort, we decided to look more closely at individual households of cats, either in pairs or trios. We sampled ten multi-cat households from our dataset and compared viral RNA levels among members of that household (Figure 2). Results from these comparisons agree with our data from the entire study cohort. Within individual households, there are significant differences between individual sample types. In some households, the FIP cats had significantly higher viral RNA, while in others, the housemates had higher levels of viral RNA (Figure 2). For example, fecal swab viral RNA levels in Household 1 are significantly higher in the healthy housemate than in the FIP cat. However, in Household 2, the cat with FIP had significantly higher amounts of viral RNA in its fecal swab sample. Interestingly, in a single household (Household 10), the healthy housemate had significantly higher amounts of viral RNA in each sample type. Given the unknown infection history of the sampled cats, we can make no conclusions as to the timing or progression of FCoV infections in individual households.

### 3.4. RNA Copies Do Not Differ Between Sexes and Cluster in Young Cats

Using quantifiable samples from each sample type, we next compared the amounts of viral RNA between male and female cats. We found that 63.5% of cats in our study with at least one quantifiable sample were male (36.5% female). Out of all the cats with FIP, 61.1% were male (38.9% female). In healthy cats with at least one quantifiable sample, 65.5% were male (34.5% female). In every sample type, there appeared to be no statistical difference in the amount of viral RNA observed (Appendix A). We then compared our results to the ages of the cats at time of presentation to the clinic. Young cats were more likely to test with quantifiable levels in all sample types (Appendix A). The median age of cats with at least one quantifiable sample was 7.8 months. Housemates with at least one quantifiable sample tended to be older than cats with FIP, with a median age of 1.59 years compared to 0.66 years with FIP cats (Appendix A).

## 4. Discussion

This study shows that both cats clinically suspected of FIP as well as their healthy housemates had detectable levels of FCoV RNA in various sample types. The low-to-moderate levels of RNA observed in the blood (both plasma and whole blood) as well as the higher amounts of FCoV RNA isolated from fecal swabs, conjunctival swabs, and effusions agree with past studies showing varying levels of viral RNA in cats with and without clinical FIP [18,42,43,44,45,46]. Our study uses the Taqman FAM/ZEN/IABQ probe-based, two-step RT-qPCR system to target a conserved sequence between primer binding sites. While single-quenched probes have been developed [46,47], to our knowledge, a dual-quenched probe has not been used before to study clinical samples of feline coronaviruses. Dual-quenched probes have been shown to offer increased sensitivity in other qPCR assays [51,52,53]. Previously published assays to detect FCoV RNA were either non-quantitative or only reported cycle thresholds (Cts) [16,42,43,45,47,50,54]. Cts in our assay were lower than those previously reported for whole blood and plasma samples [29,43] and approximately the same for effusions and fecal swabs [39,43,46]. One study by Pedersen et al. reported an LoD of ~1000 RNA copies/mL of blood, which is close to the LoD presented here [29]. Another publication by Meli et al. had a slightly higher amount of viral RNA in the effusions of FCoV-infected cats, but with a higher limit of detection for their assay (~1850 copies/mL of blood or ~20,000 copies/g feces) [39]. We conclude here that the sensitivity of our assay is the same or better than previously published results, even if the amount of viral RNA detected in our cohorts does not differ significantly from previous reports.

The assays here use a standard curve based on the absolute quantification of RNA copies. Such methods allow the quantification and comparison of different sample types across different cats. However, owing to sampling from client-owned cats, we acknowledge that the timing of viral infection differs between cats sampled in this study. All time points used in the present study are when the FIP cats presented to the clinic. Those cats who are very early or much later in the course of viral infection may have had undetectable levels of virus at some sampling sites due to virus clearance or low viral replication.

In our cohort of cats, the levels of viral RNA did not differ significantly between cats who were suspected of FIP and their housemates, indicating that detecting viral RNA is not enough to conclusively diagnose FIP. In unexposed, healthy cats with unknown infection histories, we observed a much lower percentage of quantifiable RNA in all samples compared to the feline housemate group (Table 1 and Appendix A). The RT-qPCR assay presented here, like all viral qPCRs, does not differentiate between actively replicating virus and residual viral RNA. However, we do note that we were able to detect FCoV RNA in the blood (whole blood and plasma) of cats that did not have FIP. Such findings lead us to confirm that systemic spread of FCoV alone is not sufficient to cause FIP, agreeing with several other studies [45,46,55,56]. Also, we were able to detect high levels of viral RNA in the feces of both FIP cats and their housemates. Generally, we quantified higher amounts of feline coronavirus RNA derived from the feces than other sample types. We speculate that the presence of FCoV RNA in these fecal samples may suggest that shedding of FCoV is occurring in both FIP cats and their housemates. Based on the overall rate of positivity and lack of difference in viral RNA amounts between FIP cats and their healthy housemates, we believe that both groups are shedding approximately equal amounts of virus. Other studies have observed the same amounts of viral RNA in the feces when assessing viral shedding [39,43,46,57,58]. This study did not assess the viral genomic populations that may be shed in the feces and whether the virus shed from FIP and healthy cats contains spike mutants associated with systemic spread.

Uniquely, we also observed the presence of FCoV viral RNA in conjunctival swabs from both FIP cats and their healthy housemates (Figure 1D). While other research has detected the presence of FIP-associated FCoV RNA in the eyes of diseased cats [18,25], we show that FCoV viral RNA is also detectable to the same degree in healthy cats. Furthermore, we observed detectable amounts of viral RNA in all nine cats’ conjunctiva without known FCoV exposures. Generally, conjunctival disease is common in cats and may be caused by a range of infectious agents [33]. Most notable of these are feline herpes virus (FHV), feline immunodeficiency virus (FIV), and feline calicivirus (FCV) [33]. An NCBI BLAST (https://blast.ncbi.nlm.nih.gov/Blast.cgi, accessed on 30 June 2025) search for the FCoV primers used in this study shows no match to any known sequence of FHV, FIV, or FCV, leading us to conclude that our detected RNA levels do not arise from another common cause of feline conjunctival disease, including in the nine unexposed cats. Understanding FCoV infection of feline conjunctiva may help shed light on the types of tropism changes that contribute to FCoV-associated disease.

Cats were recruited with the hypothesis that infections in multi-cat households stem from a single founder virus, which developed into FIP in some cats but not in their housemates. The high potential for relatedness among infecting viruses within a household is a controlling factor allowing finer comparisons between FIP and non-FIP pathotype viruses. However, without high-confidence sequencing of FCoV genomes derived from these samples, we cannot know how related the housemate-infecting virus is to the FIP-causing virus in a household. In this case study, we note the same trends among cats from the same household as in the aggregated cohort of cats. However, there are no correlations in viral RNA levels with FIP disease when examining individual multi-cat households. Furthermore, there were no correlations in viral RNA between the sexes. Overall, we recruited more males in our study than females, but the amount of viral RNA did not differ for any sample type. However, cats with quantifiable viral RNA did tend to be younger, agreeing with past research illustrating that FCoV is more prominent in younger cats [14,15]. In our study, healthy housemate cats who had quantifiable virus in at least one sample tended to be older than FIP cats but still under 2 years of age.

We acknowledge that our assay may detect CCoV or PRCV (Appendix A). However, we can identify no reported cases of a natural CCoV or PRCV infection of cats, likely due to a lack of robust surveillance. Under experimental conditions, it has been shown that CCoV and another related porcine virus (transmissible gastroenteritis virus, TGEV) are able to infect feline hosts [59,60,61,62]. Additionally, some serotypes of feline coronavirus arose from a recombination event with CCoVs, illustrating that co-infection is plausible [63].

When assessing the possibility of detecting something other than FCoV, we used our RT-qPCR assay on cats with no known exposure to FCoV. Most of the samples from the nine cats we sampled had RNA levels below the limit of detection. When considering the positive values from this group, the level of positivity is lower than what has been previously reported in healthy housecats [50,64].

The RT-qPCR assay presented here uses a two-step protocol, consisting of a reverse transcription reaction independently of the qPCR. The major advantage of this system is to be able to generate a reserve stock of FCoV cDNA from each sample for future experiments, such as FCoV sequencing. One-step RT-qPCR systems may offer increases in sensitivity, when sampling from sources of coronavirus shedding [65,66], especially for rarer RNA species [67]. A two-step methodology has also been shown to reduce the number of false negatives [66]. The methods presented in this case study could easily be adapted into a one-step system.

We conclude that the development of FIP does not correlate with the amount, nor the anatomical location of viral RNA, agreeing with what others have published [45,46,54,55,56]. Thus, the exact reasons for the emergence of FIP within cats remain unknown. However, the methodologies presented in this case study illustrate a new way of looking at feline coronavirus infections in the context of natural infection by directly comparing different sample types from two cohorts of cats: those with clinically suspected FIP and their housemates, by using a highly sensitive, dual-quenched Taqman-probe-based qPCR assay. Additional analyses using viral genome sequencing may shed light on FCoV intrahost evolution and the diversity of viral populations that arise before and after FIP diagnosis. Additionally, biochemical examinations of both FCoV and FIP-associated FCoV and how they may interact differently with the feline immune system are needed to fully understand FIP pathogenesis.

## Figures and Tables

**Figure 1 viruses-17-00948-f001:**
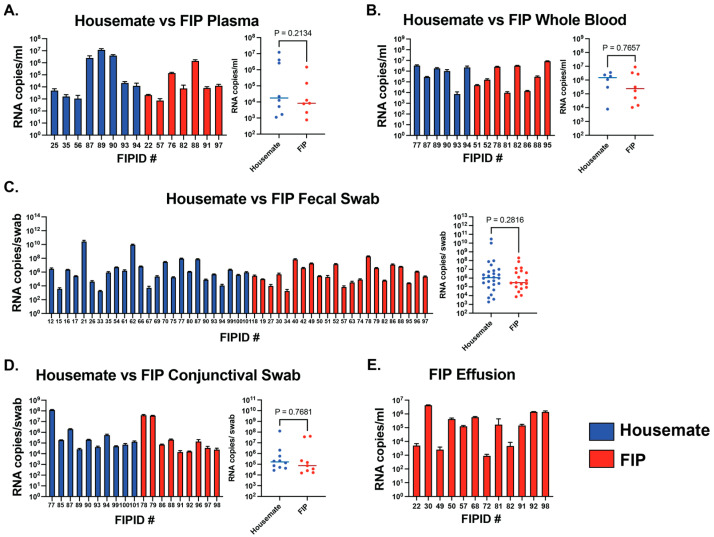
Quantifiable RT-qPCR data as RNA copies/mL or RNA copies/swab in each sample type for each FIP identification number (FIPID #): (**A**) plasma; (**B**) whole blood; (**C**) fecal swabs; (**D**) conjunctival swabs; and (**E**) effusion. Samples that were detected but not quantifiable or were below the limit of detection are not shown. Unpaired *t*-test results are displayed as *p*-values. Statistical significance was defined as *p* < 0.05.

**Figure 2 viruses-17-00948-f002:**
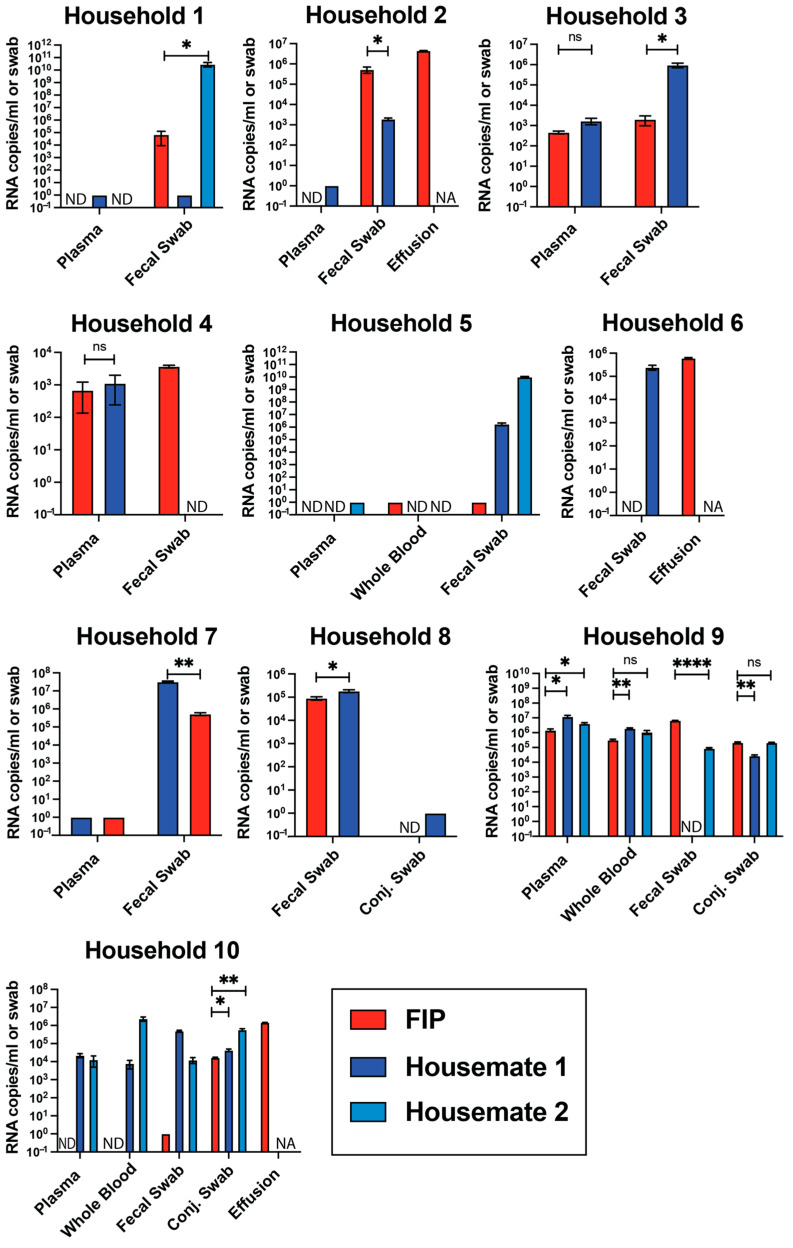
Comparisons between FIP cats and their housemates in selected households. Households selected based on RT-qPCR results. Samples detectable but not quantifiable were given a value of 1 and excluded from statistical analysis. Samples below the limit of detection were given a value of 0 and labeled as not detected. NA, not applicable; ND, not detected; ns, not significant; * = *p* < 0.05; ** = *p* < 0.01; *** = *p* < 0.001; **** = *p* < 0.0001 based on paired *t*-tests.

**Table 1 viruses-17-00948-t001:** Collected data from our cohort displayed as percent positivity in all cats. Data was sorted into quantifiable, detected, or below the limit of detection for each sample type.

	Plasma (68)	Whole Blood (40)	Fecal Swab (65)	Conjunctival Swab (28)	Effusion (16)
% Quantifiable (n)	22.06% (15)	35.00% (14)	72.58% (48)	67.86% (19)	75.00% (12)
% Detected (n)	19.12% (13)	15.00% (6)	8.06% (5)	10.71% (3)	0.00% (0)
% Below the LoD (n)	58.82% (40)	50.00% (20)	19.36% (12)	21.43% (6)	25.00% (4)

## Data Availability

The data presented within the scope of this study can be provided by the corresponding author upon request. Clinical data for recruited cats is publicly available on Dryad at https://doi.org/10.5061/dryad.83bk3jb51.

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
