# Peer review of "Detection of Feline Coronavirus RNA in Cats with Feline Infectious Peritonitis and Their Housemates"

_viruses, 2025, doi:10.3390/v17070948_

Round 1

Reviewer 1 Report

Comments and Suggestions for Authors

This study describes a PCR assay for detection of FECV/FIPV in plasma/whole blood/conjunctiva/feces/effusions in cats.

Introduction:

Discuss what is known about FECV/FIPV tropism in vivo (FECV : intestinal epithelial cells vs. FIPV : macrophages). Please add references about FECV/FIPV tropism.

Indicate why you assumed that cats in the same household harbor the same parental FECV. It is possible that littermates (infected with FECV very shortly after birth) harbor the same parental virus but housemates may not. It is thought that nearly all cats harbor FECV. Their initial infection may protect them from re-infection or dual infections. So, they may or may not acquire additional variants of FECV as adults when exposed to other cats.

M&M:

Lines 82-84. These two sentences seem redundant. Were there other alternative diagnoses that lead to exclusion of cats from this study?

Results:

The results from the conjunctival swabs are interesting. Do you know if your primers might amplify other viruses that are commonly associated with conjunctival disease in cats (e.g., calicivirus, herpesvirus)? [A quick BLAST search will likely ensure this is not the case.] Please add a short comment in the discussion about conjunctival diseases in cats.

Additional comparisons of viral load that might be very interesting could include: age of sampling, co-morbidities cats are experiencing, other stressors in the cats' lifestyle, spay/neuter status of the animals, male vs. female...

Discussion:

Discussion of viral tropism, known genetic differences between FECV and FIPV, and other viral causes of conjunctivitis in cats would be beneficial.

I highly recommend additional collaborations/consults with veterinary virologists/pathologists. There is information missing about the current understanding of FECV/FIPV pathogenesis that should be added to the paper. A veterinarian virologist/pathologist also would be able to contribute to selecting subsets of animals in this study that might reveal significant findings that are not reported in this manuscript.

Author Response

Comment 1: Discuss what is known about FECV/FIPV tropism in vivo (FECV : intestinal epithelial cells vs. FIPV : macrophages). Please add references about FECV/FIPV tropism.

Response 1: We have added more detail about what is known about FCoV tropism in the introduction, lines 49 to 62. This information includes a brief synopsis of cell type tropisms and how these tropisms are hypothesized to contribute to FIP disease. We have also included information on genetic differences speculated to contribute to systemic spread of FCoV as well as FIP pathogenesis.

Comment 2: Indicate why you assumed that cats in the same household harbor the same parental FECV. It is possible that littermates (infected with FECV very shortly after birth) harbor the same parental virus but housemates may not. It is thought that nearly all cats harbor FECV. Their initial infection may protect them from re-infection or dual infections. So, they may or may not acquire additional variants of FECV as adults when exposed to other cats.

Response 2: You make a good point here. We made the assumption that both groups’ viruses are related based on close proximity between the housemate and the diseased cat in each household. We believe that is it likely that viruses transmit between the two, but without sequencing data there is no way to know the relatedness of the viruses. To make this more clear, the word “assumption was added to line 110 to replace “hypothesis” and a short comment was added to the discussion in lines 409-412 to highlight the relatedness of the viruses can be determined by sequencing.

Comment 3: Lines 82-84. These two sentences seem redundant. Were there other alternative diagnoses that lead to exclusion of cats from this study?

Response 3: We agree that those two sentences are redundant and have deleted the first sentence. In order to clarify, we have added a comment in lines 126-129 explaining that each case was monitored by a board-certified vet who made each diagnoses. Alternative diagnoses that lead to exclusion of cats from the study are listed in lines 127-130.

Comment 4: The results from the conjunctival swabs are interesting. Do you know if your primers might amplify other viruses that are commonly associated with conjunctival disease in cats (e.g., calicivirus, herpesvirus)? [A quick BLAST search will likely ensure this is not the case.] Please add a short comment in the discussion about conjunctival diseases in cats.

Response 4: An NCBI blast search using the sequence of the primers and probe used in the study was done during study design. There are no known non-coronavirus sequences that match the sequences of our primers and probe. In order to describe this better, we have added lines 391-404 in the discussion section describing causes of conjunctival diseases in cats and the binding specificity of our primers.

Comment 5: Additional comparisons of viral load that might be very interesting could include: age of sampling, co-morbidities cats are experiencing, other stressors in the cats' lifestyle, spay/neuter status of the animals, male vs. female...

Response 5: Thank you for this comment. We looked at both age and sex of cats with at least one quantifiable sample and did not observe any differences. Younger cats tended to be more positive, which has previously been reported in literature. This data is summarized in supplemental figures 2 and 3. We also added a paragraph to the results section to explain this data in lines 330-340. Additional demographic data such as spay/neuter status and breed will be uploaded to the Dryad data sharing site.

Comment 6: Discussion of viral tropism, known genetic differences between FECV and FIPV, and other viral causes of conjunctivitis in cats would be beneficial.

Response 6: We agree that the manuscript would be improved by a more in-depth discussion of FECV and FIPV and conjunctivitis in cats. We believe our response to comments 1 and 4 of this reviewer have satisfied this comment.

Comment 7: I highly recommend additional collaborations/consults with veterinary virologists/pathologists. There is information missing about the current understanding of FECV/FIPV pathogenesis that should be added to the paper. A veterinarian virologist/pathologist also would be able to contribute to selecting subsets of animals in this study that might reveal significant findings that are not reported in this manuscript.

Response 7: Thank you for the suggestion. As a part of responding to all reviewer comments, we have added more information regarding the pathotype switch between FECV and FIPV as well as a discussion on tropisms, antibody dependent enhancement, and viral shedding. These changes can be found throughout the manuscript. As far as selecting subsets of animals, we have added information regarding sex and age to the supplement and results.

Reviewer 2 Report

Comments and Suggestions for Authors

In this manuscript, the authors present data about FCoV RNA distribution and loads at different sampling sites (blood, conjunctiva and feces) of cats clinically diagnosed with FIP and their healthy companion animals.

While the results are interesting, they are not particularly novel, and the manuscript contains several critical issues that need to be carefully addressed.

Major points:

Throughout the manuscript, the authors refer to the use of qRT-PCR. As currently presented, this may lead readers to assume a one-step RT-qPCR was used. However, since the method actually involves a two-step process (first cDNA synthesis and then qPCR), the authors are kindly requested to clarify this point and revise the terminology accordingly throughout the manuscript to avoid any misunderstanding

Abstract:

In the abstract conclusion, the authors state there is “a need for closer examination of FCoVs.” This statement is overly vague and lacks specificity. The authors are encouraged to clarify what particular aspects of FCoVs warrant further investigation—such as pathogenesis, transmission dynamics, etc.—to provide more meaningful and informative conclusion sentence.

Introduction:

Lane 42: There are numerous hypotheses regarding the pathogenesis of FIP, so referring to an "unknown mechanism" is overly simplistic. Please provide more details or specify which aspects of the mechanism remain unclear or are under debate.

Lane 42 and 44 and entire manuscript: Please change the nomenclature throughout the manuscript according to the new guidelines of the European Advisory Board for Cat diseases (ABCD). There is no FECV or FIPV but FCoV and FIP-associated FCoV.

Lanes 55-57: The sentence “The gold standard for detection of FCoVs is to use in situ hybridization probes targeting the FCoV nucleocapsid or matrix protein in fixed tissues from biopsies or necropsies” is misleading. In situ hybridization detects viral RNA or DNA, not proteins, so referencing nucleocapsid or matrix proteins in this context is inaccurate. Additionally, the gold standard for FIP diagnosis is still considered to be immunohistochemistry, which detects viral antigens using specific antibodies in tissue lesions or biopsies. For detecting FCoV RNA, the gold standard is RT-PCR. Please revise the sentence to clearly distinguish between these diagnostic methods and their respective targets.

Lanes 66-67: Given the aim of this manuscript, the authors should provide a more thorough review and discussion of the existing literature on the detection of viral RNA in different compartments of cats with FIP as well as in FCoV-infected animals. Additionally, the manuscript overlooks numerous studies on FCoV viral shedding, which is directly relevant to the authors’ own research focus. This omission should be addressed to strengthen the context and scientific grounding of the study.

Materials &Methods

Paragraph 2.1.: please specify here how many cats for each group were enrolled in the study and which kind of data (age, breed, sex, etc.)  were recorded from these study cats.

Paragraphs 2.3. and 2.4.: How did the authors monitor for potential sample contamination and carry-over during the workflow? Were negative controls included at the stages of RNA extraction, cDNA synthesis, and PCR? Please provide this information in the relevant section of the manuscript to clarify the measures taken to ensure the reliability of the results.

Results:

Lanes: 177-179: these two additional standards are not mentioned in the M&M section, please add this information.

Paragraph 3.1.: If the authors designed the standards themselves, they should have precise knowledge of the sequences used, including the number and positions of any mismatches between the FCoV primers and probe and the CCV and PRCV sequences. These details are important for assessing specificity and should therefore be reported in the manuscript.

Mismatches between primers or probes and their target sequences can significantly impact (reduced binding efficiency, lower amplification efficiency and delayed Ct values, reduced fluorescence signal) not only the specificity but also the efficiency, and sensitivity of a qPCR assay. Please report in the text also the efficiencies of the assay with the 3 standards and also the calculated LoQs and LoDs.

Lanes 183-184: The authors are encouraged to review the relevant literature more thoroughly before making such statements. It is well-documented that CCoV can also infect cats; a notable example is the FCoV-23 strain identified in Cyprus, which originated from a recombination event between FCoV and CCoV. Furthermore, this kind of comments belongs to the discussion section.

Lanes 193-194: this comment should be moved to the discussion section

Paragraph 3.2: the authors may consider adding in each subchapter the number of positive and negative animals in each subgroup.

Table 1: It would be informative to break down the percentages by the two groups (FIP and housemates), except for effusions, which are not present in the healthy housemates (hopefully).

Lane 239: How many of the cats with FIP had effusions?

Lanes 250-252: The sentence is unclear as currently written; please revise it to improve clarity.

Figure 2: Why was no statistical analysis performed for the fecal swabs in household 5, one of the fecal swabs of household 1, plasma of the household 7 and fecal swabs of household 10? Please clarify.

Discussion:

Overall, the discussion is limited and would benefit from a more thorough comparison of the results with findings from similar studies in the existing literature.

Lanes 270-271: The authors are encouraged to compare the sensitivity of their assay with that of other established FCoV RT-qPCR methods, providing specific examples. Additionally, it would be valuable to discuss the potential advantages and disadvantages of using a two-step protocol versus a one-step approach.

Lanes 293-294: Numerous studies have been published on fecal FCoV shedding. The authors are encouraged to compare their findings with those reported in the existing literature to provide context and highlight any similarities or differences.

Lanes 294-295: This statement is unclear. The authors are encouraged to clarify their intended meaning—specifically, what is meant by "assessing viral populations"? Please elaborate to ensure the concept is clearly understood by the reader.

Certain results, such as the unusual case of household 10, are not addressed in the discussion. Overall, the authors should provide a more detailed analysis and interpretation of their findings to strengthen the discussion section.

Lanes 303.304: This is an important statement that appears to contradict findings from other studies. The authors should cite the relevant literature in this context and highlight both the similarities and discrepancies to provide a balanced and well-supported discussion.

Minor points:

Lanes 17-18: “infected FIP cats” consider changing to “FCoV-infected cats with FIP”

Lanes 35-38: provide references to these statements

Lane 88: how much later?

Lane 121: what about conjunctival swabs?

Lanes 123-125: was the carrier RNA added also to the whole blood samples?

Lanes 152-166: what about the thermal conditions for the PCR?

Lane 180: there is a typo in the legend of the supplementary Figure 1: “sensitivity” and not “sensativity”

Lane 186: change “DNA” to “genomic DNA”

Lane 191: It is supplementary table 1 and not supplementary Figure 1

Figure 1: please remove the “+” sign next to the “FIP” in the figure legend. It is just FIP and Housemate.

Table1: please check: the sum of the percentages for fecal swabs is not 100%

Figure 2: please remove the “+” sign next to the “FIP” in the figure legend. It is just FIP and Housemate.

Lane 263: please add healthy before “housemates”

Author Response

Comment 1: Throughout the manuscript, the authors refer to the use of qRT-PCR. As currently presented, this may lead readers to assume a one-step RT-qPCR was used. However, since the method actually involves a two-step process (first cDNA synthesis and then qPCR), the authors are kindly requested to clarify this point and revise the terminology accordingly throughout the manuscript to avoid any misunderstanding

Response 1: Thank you for pointing this out. We have changed “qRT-PCR” to “RT-qPCR” throughout the manuscript to specify when the quantification is occurring. Additionally, to limit any confusion, we have added the “two-step” description numerously throughout the manuscript to describe our assay.

Comment 2: In the abstract conclusion, the authors state there is “a need for closer examination of FCoVs.” This statement is overly vague and lacks specificity. The authors are encouraged to clarify what particular aspects of FCoVs warrant further investigation—such as pathogenesis, transmission dynamics, etc.—to provide more meaningful and informative conclusion sentence.

Response 2: This is an excellent point. The manuscript has been updated (lines 20-24) to clarify the need for more research on viral pathogenesis independent from qPCR-based studies on viral tropism.

Comment 3: Lane 42: There are numerous hypotheses regarding the pathogenesis of FIP, so referring to an "unknown mechanism" is overly simplistic. Please provide more details or specify which aspects of the mechanism remain unclear or are under debate.

Response 3: The authors agree with the point and thank the reviewer for bringing it up. We have updated the introduction to include what is known about FCoV tropism and how FIP may arise. We specifically mention antibody dependent enhancement and viral mutations that have been hypothesized to be involved with the pathogenesis of FIP. This information is contained in lines 49-67.

Comment 4: Lane 42 and 44 and entire manuscript: Please change the nomenclature throughout the manuscript according to the new guidelines of the European Advisory Board for Cat diseases (ABCD). There is no FECV or FIPV but FCoV and FIP-associated FCoV.

Response 4: The change from FECV and FIPV to FCoV and FIP-associated FCoV has been made throughout the whole manuscript. However because “FECV” and “FIPV” are used historically in FCoV literature, we have retained “FECV” and “FIPV” as keywords for the manuscript.

Comment 5: Lanes 55-57: The sentence “The gold standard for detection of FCoVs is to use in situ hybridization probes targeting the FCoV nucleocapsid or matrix protein in fixed tissues from biopsies or necropsies” is misleading. In situ hybridization detects viral RNA or DNA, not proteins, so referencing nucleocapsid or matrix proteins in this context is inaccurate. Additionally, the gold standard for FIP diagnosis is still considered to be immunohistochemistry, which detects viral antigens using specific antibodies in tissue lesions or biopsies. For detecting FCoV RNA, the gold standard is RT-PCR. Please revise the sentence to clearly distinguish between these diagnostic methods and their respective targets.

Response 5: Thank you for correcting this. Terminology between IHC and hybridization was switched. The manuscript has been updated (lines 77-80) to correctly describe IHC as targeting proteins and RT-PCR as targeting RNA.

Comment 6: Lanes 66-67: Given the aim of this manuscript, the authors should provide a more thorough review and discussion of the existing literature on the detection of viral RNA in different compartments of cats with FIP as well as in FCoV-infected animals. Additionally, the manuscript overlooks numerous studies on FCoV viral shedding, which is directly relevant to the authors’ own research focus. This omission should be addressed to strengthen the context and scientific grounding of the study.

Response 6: The authors agree with this comment. We have added a section in the introduction (lines 86-101) to discuss what is known about the amount of viral RNA in various tissues and body fluids as well a brief explanation of fecal shedding in both sick and healthy cats.

Comment 7: Paragraph 2.1.: please specify here how many cats for each group were enrolled in the study and which kind of data (age, breed, sex, etc.)  were recorded from these study cats.

Response 7: The numbers of cats included were added throughout the materials and methods section, including total number of cats (line 117) how many cats had FIP and how many were healthy housemates (lines 132-133). The types of data collected from the cohort were also listed in lines 124-127.

Comment 8: Paragraphs 2.3. and 2.4.: How did the authors monitor for potential sample contamination and carry-over during the workflow? Were negative controls included at the stages of RNA extraction, cDNA synthesis, and PCR? Please provide this information in the relevant section of the manuscript to clarify the measures taken to ensure the reliability of the results.

Response 8: Negative controls for RNA extraction include extraction from two feline cat cell lines for FCoV RNA . We additionally included RNA extractions from samples from healthy cats not co-habitating with a FIP cat (Line 248-252). Because client-owned cats have unknown infection histories, it is difficult to conclude that these healthy cats have not previously been exposed to FCoV so it is difficult to rely on them as true negatives. Indeed many of these samples are positive for FCoV (Line 247-251) though with less positivity than healthy cats co-habitating with a FIP cat. Controls for cDNA synthesis and qPCR include the inclusion of no-sample/template controls with Ct values > 40 showing a lack of contamination crossover among samples (210-211). These methods give us confidence that our assay is specific and cross contamination through reagents was negligible.

Comment 9: Lanes: 177-179: these two additional standards are not mentioned in the M&M section, please add this information.

Response 9: Thank you for pointing out the omission. The two standards have been added to the M&M section (lines 180-181). The PRCV and CCoV standards were produced in the same way as the FCoV standard.

Comment 10: Paragraph 3.1.: If the authors designed the standards themselves, they should have precise knowledge of the sequences used, including the number and positions of any mismatches between the FCoV primers and probe and the CCV and PRCV sequences. These details are important for assessing specificity and should therefore be reported in the manuscript.

Response 10: The primer and probe sequences for our FCoV primers are also found in CCoV and PRCV genomes (100% match). Details about the locations to matched sequences has been added (lines 233-235)

Comment 11: Mismatches between primers or probes and their target sequences can significantly impact (reduced binding efficiency, lower amplification efficiency and delayed Ct values, reduced fluorescence signal) not only the specificity but also the efficiency, and sensitivity of a qPCR assay. Please report in the text also the efficiencies of the assay with the 3 standards and also the calculated LoQs and LoDs.

Response 11: Thank you for bringing this point up. The efficiencies for the CCoV and PRCV assays were lower than FCoV (even though the primer probe sequences matched). Cts for CCoV were approximately the same at dilute levels, so the LoQ was the same as the FCoV curve. For PRCV, it was about two logs higher. This is now reported in the manuscript in lines 236-242.

Comment 12: Lanes 183-184: The authors are encouraged to review the relevant literature more thoroughly before making such statements. It is well-documented that CCoV can also infect cats; a notable example is the FCoV-23 strain identified in Cyprus, which originated from a recombination event between FCoV and CCoV. Furthermore, this kind of comments belongs to the discussion section.

Response 12: We have removed the statement. You are correct that FCoV-23 (as well as other serotype II viruses) arose from a recombination event between FCoV and CCoV. However, it is not known where that recombination event occurred (dog, cat, or another animal) and there has been no direct biochemical evidence of natural CCoV infection in feline hosts. Under experimental conditions, it’s been observed that CCoV can infect cats and use feline cell receptors, suggesting that natural CCoV infection is indeed plausible. In order to clarify this point, we have added a description of CCoV and FCoV coinfection in the discussion (lines 422-427)

Comment 13: Lines 193-194: this comment should be moved to the discussion section

Response 13: The comment has been moved to the discussion section (lines 432-433)

Comment 14: Paragraph 3.2: the authors may consider adding in each subchapter the number of positive and negative animals in each subgroup.

Response 14: Thank you for pointing this out. We categorize our results into “quantifiable”, “detected”, “below LoD”, and “undetected” to better describe the qPCR values observed. In this way, one could view “quantifiable” samples as positive and “below LoD” and “undetected” as negative. The n’s of each sample has been added to Table 1 to indicate group size. In order to illustrate the statistics in Figure 1, we only use quantifiable data. The n’s for each sample type between FIP and housemate cats as well as the percent positivity (of quantifiable data) has been added to each paragraph in the results.

Comment 15: Table 1: It would be informative to break down the percentages by the two groups (FIP and housemates), except for effusions, which are not present in the healthy housemates (hopefully).

Response 15: This is a good point. For each sample type at each quantification level, the amount of FIP vs housemate viral RNA was approximately equal or had low n’s. The total amount of cats in each quantification level added to Table 1 and percent positivity between FIP and housemates were added to each paragraph.

Comment 16: Lane 239: How many of the cats with FIP had effusions?

Response 16: There were 16 cats with effusions. This information has been added to Table 1.

Comment 17: Lanes 250-252: The sentence is unclear as currently written; please revise it to improve clarity.

Response 17: Thank you for the feedback. We meant that statement to describe the lack of any discernable patterns between individual households. We have clarified that point in the manuscript in lines 312-317.

Comment 18: Figure 2: Why was no statistical analysis performed for the fecal swabs in household 5, one of the fecal swabs of household 1, plasma of the household 7 and fecal swabs of household 10? Please clarify.

Response 18: For the purpose of illustrating detected but non-quantifiable samples, we gave the value of 1 to detected samples in these graphs. However, these weren’t used in statistical analysis because there was no quantifiable qPCR value associated with that sample. This was stated in the original manuscript both in the figure legend and in the methods section talking about statistics. In order to clarify, we have added a statement in the legend of figure 2 stating that we excluded those samples from the statistical analysis.

Comment 19: Overall, the discussion is limited and would benefit from a more thorough comparison of the results with findings from similar studies in the existing literature.

Response 19: Thank you for the comment. We are hoping that our responses to comments 20-24 satisfy this point.

Comment 20: Lanes 270-271: The authors are encouraged to compare the sensitivity of their assay with that of other established FCoV RT-qPCR methods, providing specific examples. Additionally, it would be valuable to discuss the potential advantages and disadvantages of using a two-step protocol versus a one-step approach.

Response 20: Thank you for the suggestion. We have examined the literature and made comparisons between Ct values and LoDs to assess sensitivity. This is summarized in lines 353-363. Additionally, we have added a brief discussion of the strengths of one- vs two- step RT-qPCR in lines 435-442.

Comment 21: Lanes 293-294: Numerous studies have been published on fecal FCoV shedding. The authors are encouraged to compare their findings with those reported in the existing literature to provide context and highlight any similarities or differences.

Response 21: As a part of our response to comment 20, we stated that the amount of viral RNA we observed in the feces was approximately the same as what others have published. To address this comment, we added a comment about shedding in lines 384-388.

Comment 22: Lanes 294-295: This statement is unclear. The authors are encouraged to clarify their intended meaning—specifically, what is meant by "assessing viral populations"? Please elaborate to ensure the concept is clearly understood by the reader.

Response 22: We have expanded on this idea in lines 453-456.

Comment 23: Certain results, such as the unusual case of household 10, are not addressed in the discussion. Overall, the authors should provide a more detailed analysis and interpretation of their findings to strengthen the discussion section.

Response 23: Thank you for the comment. We agree that we can provide a more in-depth discussion about our findings and have added information to the discussion comparing our results to others (lines 351-361). We have added lines 317-319 highlighting Household 10 in the results section.

Comment 24: Lanes 303.304: This is an important statement that appears to contradict findings from other studies. The authors should cite the relevant literature in this context and highlight both the similarities and discrepancies to provide a balanced and well-supported discussion.

Response 24: Several studies have pointed to the presence of FCoV RNA in several sampling sites from cats with and without FIP, and ours improves on the methodology they have developed. A comment about the complications of FIP pathogenesis and how our assay improved on FCoV detection are included in lines 445-447 and 453-456 respectively.

Minor points:

Comment 25: Lanes 17-18: “infected FIP cats” consider changing to “FCoV-infected cats with FIP”

Response 25: Thank you for pointing this out. The manuscript has been updated accordingly.

Comment 26: Lanes 35-38: provide references to these statements

Response 26: The manuscript has been updated accordingly by adding references.

Comment 27: Lane 88: how much later?

Response 27:  This is a good question. Unfortunately, we are sampling from client-owned cats in this study. This means the scheduling of sample collection is dependent on when the cat owners are available to bring their healthy cats to the vet. Typically, we were able to sample from housemates within a week. The manuscript has been updated in lines 133-136 to reflect this.

Comment 28: Lane 121: what about conjunctival swabs?

Response 28: Thank you for pointing this out. The manuscript has been updated to add conjunctival swabs in the description in line 168.

Comment 29: Lanes 123-125: was the carrier RNA added also to the whole blood samples?

Response 29: Yes, carrier RNA was added to every sample. We clarified this point in line 173.

Comment 30: Lanes 152-166: what about the thermal conditions for the PCR?

Response 30: Cycling conditions for the qPCR were added in lines 205-207.

Comment 31: Lane 180: there is a typo in the legend of the supplementary Figure 1: “sensitivity” and not “sensativity”

Response 31: Thank you for pointing that out. The typo has been fixed.

Comment 32: Lane 186: change “DNA” to “genomic DNA”

Response 32: The change has been made. The line is now 245.

Comment 33: Lane 191: It is supplementary table 1 and not supplementary Figure 1

Response 33: Thank you for catching that. The manuscript has been updated accordingly. Line 250.

Comment 34: Figure 1: please remove the “+” sign next to the “FIP” in the figure legend. It is just FIP and Housemate.

Response 34: This change has been made.

Comment 35: Table1: please check: the sum of the percentages for fecal swabs is not 100%

Response 35: Thank you for pointing that out. There was an error rounding in the table that has been fixed.

Comment 36: Figure 2: please remove the “+” sign next to the “FIP” in the figure legend. It is just FIP and Housemate.

Response 36: This change has been made.

Comment 37: Lane 263: please add healthy before “housemates”

Response 37: The word “healthy” was added for clarity. Line 343

Round 2

Reviewer 1 Report

Comments and Suggestions for Authors

Version 2 of the manuscript is much improved.

Please revise Table 1 to more explicitly explain the difference between samples labeled "below the limit of detection" and samples labeled "undetected". It is unclear what the difference between those two groups is.

Please explain how your data "... add to the evidence that FIP pathogenesis is more complicated than a simple change in viral tropism..." or remove lines 444-446 from the discussion. The study design you present in this manuscript does not address the issue of FIP viral tropism.

Author Response

Comment 1: Please revise Table 1 to more explicitly explain the difference between samples labeled "below the limit of detection" and samples labeled "undetected". It is unclear what the difference between those two groups is.

Response 1: In the original manuscript, we differentiated between “below limit of detection” and “undetected” to illustrate values below the last detectable standard and values with Cts >40. You are correct that this distinction was not clear in the current manuscript. In order to address this, we combined these two groups in Table 1 in the “below limit of detection” row. The percentages and n’s have been updated.

Comment 2: Please explain how your data "... add to the evidence that FIP pathogenesis is more complicated than a simple change in viral tropism..." or remove lines 444-446 from the discussion. The study design you present in this manuscript does not address the issue of FIP viral tropism.

Response 2: You are correct that the study does not directly address viral tropisms. We have removed lines 444-446 in the discussion. Thank you for your valuable feedback on the manuscript.

Reviewer 2 Report

Comments and Suggestions for Authors

No further revisions are recommended. The revised manuscript is now acceptable for publication.

Author Response

Comment 1: No further revisions are recommended. The revised manuscript is now acceptable for publication.

Response 1: Thank you for your valuable feedback on the manuscript.